# A long non-coding RNA is required for targeting centromeric protein A to the human centromere

**Delphine Quénet, Yamini Dalal\***

Laboratory of Receptor Biology and Gene Expression, Center for Cancer Research, National Cancer Institute, Bethesda, United States

**Abstract** The centromere is a specialized chromatin region marked by the histone H3 variant CENP-A. Although active centromeric transcription has been documented for over a decade, the role of centromeric transcription or transcripts has been elusive. Here, we report that centromeric α-satellite transcription is dependent on RNA Polymerase II and occurs at late mitosis into early G1, concurrent with the timing of new CENP-A assembly. Inhibition of RNA Polymerase II-dependent transcription abrogates the recruitment of CENP-A and its chaperone HJURP to native human centromeres. Biochemical characterization of CENP-A associated RNAs reveals a 1.3 kb molecule that originates from centromeres, which physically interacts with the soluble pre-assembly HJURP/CENP-A complex in vivo, and whose down-regulation leads to the loss of CENP-A and HJURP at centromeres. This study describes a novel function for human centromeric long non-coding RNAs in the recruitment of HJURP and CENP-A, implicating RNA-based chaperone targeting in histone variant assembly.

DOI: https://doi.org/10.7554/eLife.03254.001

## Introduction

Specialized chromatin domains called centromeres play an essential role in chromosome segregation, serving as a platform for kinetochore complex formation, which in turn binds spindle microtubules at mitosis (*Verdaasdonk and Bloom, 2011*). Although centromeric DNA sequence is not uniform across species, centromere function is conserved (*Sullivan, 2009*). In humans, AT-rich 171 bp α-satellite repeats lacking any known genes are the primary DNA component of the centromere (*Waye and Willard, 1987*). Centromeres are characterized by the presence of the centromeric histone H3 variant (CENH3/CENP-A in human). Centromeric chromatin has long been considered heterochromatic, despite exhibiting a bivalent organization with heterochromatin-like post-translational modifications (PTMs), such as H3 and H4 hypo-acetylation, and transcription-coupled PTMs, including dimethylation on H3 lysine 4 (H3K4me2) (*Sullivan and Karpen, 2004*; *Heintzman et al., 2007*; *Zhou et al., 2011*). The function of such bivalent modifications has remained mysterious. Indeed, despite the common assumption that centromeres are largely dormant, a number of recent studies have pointed to the importance of transcription at centromeres in multiple organisms, which appears to be essential for the maintenance of centromere integrity (*Hall et al., 2012*). Chemical inhibition of either RNA Polymerase I (RNAPI), or RNA Polymerase II (RNAPII), results in loss of the inner kinetochore protein CENP-C, and in chromosome mis-segregation (*Wong et al., 2007*; *Chan et al., 2012*). Centromeric RNA components also seem to contribute to the structural integrity of the mitotic centromere (*Wong et al., 2007*). However, the exact timing of centromeric transcription, the polymerase involved, the identity of centromeric RNAs and their precise role in maintaining native centromere integrity in human cells has been elusive.

**\*For correspondence:**
dalaly@mail.nih.gov

**eLife digest** Before a cell divides, it copies its chromosomes. Initially, the two copies of each chromosome remain linked via their centromeres. These regions also serve as the attachment sites for the proteins that pull these two copies apart, and eventually segregate the chromosomes equally between the two newly formed cells.

Chromosome segregation is the main function of centromeres; and in most organisms, the DNA in these regions is highly repetitive and is not thought to encode any proteins. However, it has been observed that cells need enzymes called RNA polymeraseswhich transcribe stretches of DNA into RNA moleculesto be able to separate the copies of their chromosomes correctly. This suggests that RNAs transcribed from centromeres might be required for cell division, but the identity and function of these RNAs remained elusive.

Quénet and Dalal have now discovered that an RNA polymerase localizes to the DNA in human centromeres and produces RNA molecules during the early stages of the cell cycle. Two proteins– one called CENP-A and another that functions as its chaperone–that normally bind to the centromere and determine its structure were found less often in this region of the chromosome if the activity of the RNA polymerase was inhibited. Qunet and Dalal identified a specific RNA molecule that is transcribed from the centromeric DNA, which directly binds to the CENP-A protein and its chaperone before CENP-A is assembled onto the centromeric DNA. Reducing the levels of this RNA within the cells made them unable to separate their chromosomes correctly during cell divisions. Qunet and Dalal also demonstrated that this centromeric RNA is needed to specifically target both the CENP-A protein, via its chaperone, to the centromere.

The findings of Qunet and Dalal demonstrate that RNAs produced from a specific part of the chromosome can help target DNA-binding proteins back to that region's DNA sequence. Following on from this work, the next challenge will be to determine if other RNA molecules are used for the same purpose in humans and other species.

DOI: https://doi.org/10.7554/eLife.03254.002

In this study, we report that centromeric RNAs play a critical role in the maintenance of the human centromere in vivo. Using chromatin immunoprecipitation (IP), and immunofluorescence (IF) on chromatin fibers, we find that RNAPII, in conjunction with TATA-box binding protein (TBP) localizes to, and actively transcribes native human centromeres from late mitosis to early G1 (eG1). Biochemical purification and sequencing of the RNA associated with human centromeric chromatin at eG1 reveals a 1.3 kb long transcript. This RNA physically interacts with CENP-A and its chaperone HJURP (Holliday junction recognition protein) in the pre-assembly soluble state in vivo. Targeted sequence-specific knockdown of the transcript results in the formation of multipolar spindles and lagging chromosomes in subsequent mitoses, leading to chromosome instability. IF analysis of centromeric chromatin fibers demonstrates that these cellular and nuclear phenotypes arise specifically from the abrogation of CENP-A and HJURP localization to the centromere. Together, our data describe a direct physical role for a centromeric long non-coding RNA (lncRNA) in HJURP targeting, subsequent CENP-A loading, and the maintenance of centromere integrity. Our study supports the possibility that an lncRNA-based mechanism is involved in targeting CENP-A and its chaperone HJURP to the centromere.

## Results

### RNAPII is associated with native human centromeres at eG1

Centromeric transcription has been previously described in human cells, and RNAPII has been implicated in this process (*Saffery et al., 2003*; *Wong et al., 2007*; *Bergmann et al., 2011*; *Chan et al., 2012*). To investigate the timing of centromeric transcription, we used synchronized HeLa cells at G2, eG1, and G1/S to track the activated form of RNAPII (i.e., serine two phosphorylated, RNAPII$^{S2P}$) on centromeric chromatin fibers throughout the cell cycle by IF (*Figure 1—figure supplement 1*). RNAPII$^{S2P}$ co-localizes with the inner kinetochore protein CENP-B and centromeric α-satellite DNA specifically at eG1 (*Figure 1A*, *Figure 1—figure supplement 2A*). We also noted that TBP, a partner

of RNAPII normally involved in transcription initiation (*Vannini and Cramer, 2012*), is localized on eG1 CENP-A-rich fibers (*Figure 1A*). These data suggest that centromeres are actively transcribed by RNAPII machinery at eG1.

We next sought to establish whether there was a physical interaction between CENP-A chromatin and RNAPII. In order to achieve this, we extracted chromatin from non-synchronized cells after a short MNase digestion, to obtain long chromatin arrays that are rich in tri-, tetra-, and penta-nucleosomes (*Figure 1—figure supplement 2B*). From this input chromatin, centromeric chromatin was immunoprecipitated with specific antibodies against either CENP-A, or the inner kinetochore protein CENP-C or no antibody (mock IP). The mock IP control shows no enrichment of any of the centromeric proteins tested (*Figure 1—figure supplement 2C*). As expected, Western blots revealed reciprocal co-purification of CENP-A and CENP-C (*Figure 1B*, left and middle panels). RNAPII and its partner TBP also co-purified with CENP-A and CENP-C (*Figure 1B*, left and middle panels). To further establish an interaction between RNAPII and centromeric proteins, we performed the reciprocal experiment, precipitating RNAPII$^{S2P}$ from solubilized chromatin, and testing for centromeric partners. While Western blots revealed little or no interaction of RNAPII with CENP-C, a robust and

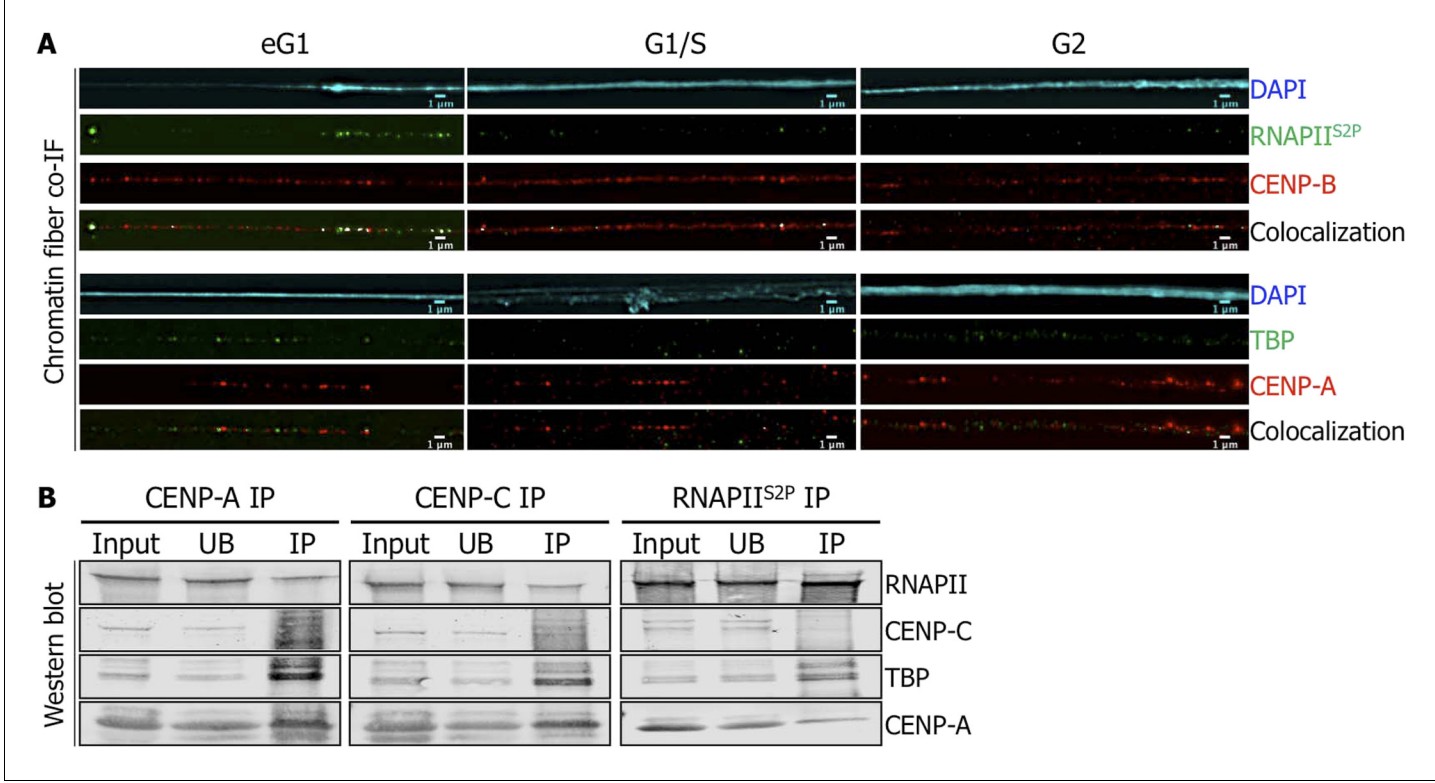

**Figure 1.** Active RNA Polymerase II (RNAPII$^{S2P}$) and TATA-box binding protein (TBP) are associated with centromeric chromatin at early G1 (eG1). (**A**) Chromatin fibers prepared from synchronized HeLa cells at eG1, G1/S, and G2 phases were co-stained for RNAPII$^{S2P}$ and TBP (green) with centromeric proteins CENP-B and CENP-A (red), respectively. The DAPI raw image is shown for a representative chromatin fiber (cyan). Three independent experiments were performed and in each, a minimum of five fibers were analyzed per slide (co-localization on the same chromatin fiber at eG1: between CENP-B and RNAPII$^{S2P}$ = 10/15; and between CENP-A and TBP = 9/16). Scale bar: 1 μm. (**B**) CENP-A, CENP-C or RNAPII$^{S2P}$ were immunoprecipitated, and co-purifying partners were detected on Western blots (1.5% of input and UB, 75% of IP). Co-IF, co-immunofluorescence; IP, immunoprecipitate; UB, unbound.

DOI: https://doi.org/10.7554/eLife.03254.003

The following figure supplements are available for figure 1:

**Figure supplement 1.** Scheme presenting the strategy for HeLa cell synchronization and treatment with drugs (actinomycin D or α-amanitin) inhibiting either RNA Polymerase I or RNA Polymerase II activity.
DOI: https://doi.org/10.7554/eLife.03254.004

**Figure supplement 2.** Active RNA Polymerase II (RNAPII$^{S2P}$) is associated with centromeric α-satellite sequences on chromatin fibers at early G1 (eG1).
DOI: https://doi.org/10.7554/eLife.03254.005

reproducible binding of RNAPII to TBP and CENP-A was observed (*Figure 1B*, right panel). Thus, active RNAPII machinery is physically associated with CENP-A chromatin at eG1.

## RNAPII transcribes native human centromeres at late mitosis-eG1

Recent studies have indicated that RNAPII transcribes centromeres at mitosis (*Chan et al., 2012*). However, our RNAPII localization data above showed RNAPII enrichment occurs primarily at eG1 (*Figure 1A*). To examine the consequence of eG1 RNAPII localization at centromeres, HeLa cells were synchronized at G2, mitosis and eG1, and briefly treated (2 hr) with drugs to specifically block either RNAPI (actinomycin D) or RNAPII (α-amanitin) activity (*Figure 1—figure supplement 1*) (*Bensaude, 2011*). After RNA extraction and retro-transcription, we determined the expression of control genes and centromeric α-satellite repeats by semi-quantitative PCR. As expected, actinomycin D and α-amanitin inhibited transcription of target genes of RNAPI (e.g., 18S rRNA) or RNAPII (e.g., GAPDH), respectively (*Figure 2*, left and middle graphs). Compared to non-treated conditions, actinomycin D treatment or α-amanitin treatment in G2 phase had no impact on centromeric α-satellite expression (*Figure 2*, right graph). Consistent with a previous study (*Chan et al., 2012*), RNAPII inhibition in mitotic cells revealed a decrease (15.6%) in centromeric transcripts (*Figure 2*, right graph). However, when RNAPII was blocked in eG1, a larger reduction (35.1%) was observed (*Figure 2*, right graph). These results suggest that RNAPII transcribes centromeres not solely at mitosis, but also throughout eG1.

## Centromeric transcription at early G1 (eG1) is required for HJURP and CENP-A targeting to the centromere

The synchrony of centromeric transcription and CENP-A recruitment onto centromeres at late mitosis-eG1 led us to examine whether active transcription is required for CENP-A loading. To test this hypothesis, we briefly treated eG1-synchronized cells with α-amanitin (2 hr) to block RNAPII activity as above, and quantified potential changes in intensity for CENP-A or CENP-B IF signal using ImageJ. Consistent with its role as a constitutive centromeric DNA-binding protein (*Verdaasdonk and Bloom, 2011*), CENP-B staining intensity was heterogeneous (*Figure 3B*, *Figure 3—figure supplement 1*), but identical in both non-treated (NT) and α-amanitin-treated cells (*Figure 3A,B*; *Supplementary file 1*), demonstrating that its localization is independent of centromeric transcription. Whereas punctate CENP-A spots can be seen under both conditions, when

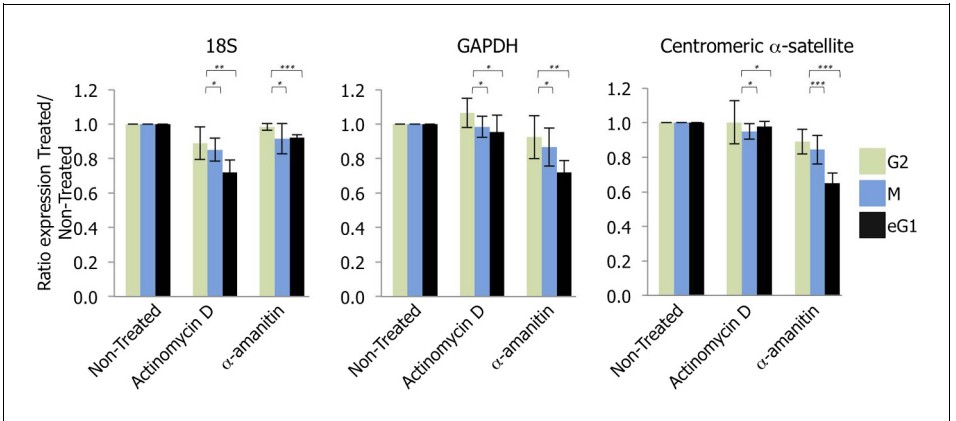

**Figure 2.** Transcription of centromeres is dependent on RNA Polymerase II (RNAPII) and occurs at early G1 (eG1). G2 (green), mitotic (M; blue), and eG1 (black) synchronized cells were treated 2 hr with either actinomycin D or α-amanitin, to block RNA Polymerase I or RNAPII, respectively. After RNA purification and retro-transcription, expression levels of control target genes (18S rRNA and GAPDH) and centromeric α-satellite transcripts were assessed by semi-quantitative PCR. The graph represents the average of three biological replicates, the y-axis plots the ratio (± SD) of gene expression after treatment (actinomycin D or α-amanitin) compared to the non-treated condition. p-values indicating statistical significance are presented where appropriate above the histograms. *p>0.1, **0.001 < p < 0.05, and ***p<0.001.

DOI: https://doi.org/10.7554/eLife.03254.006

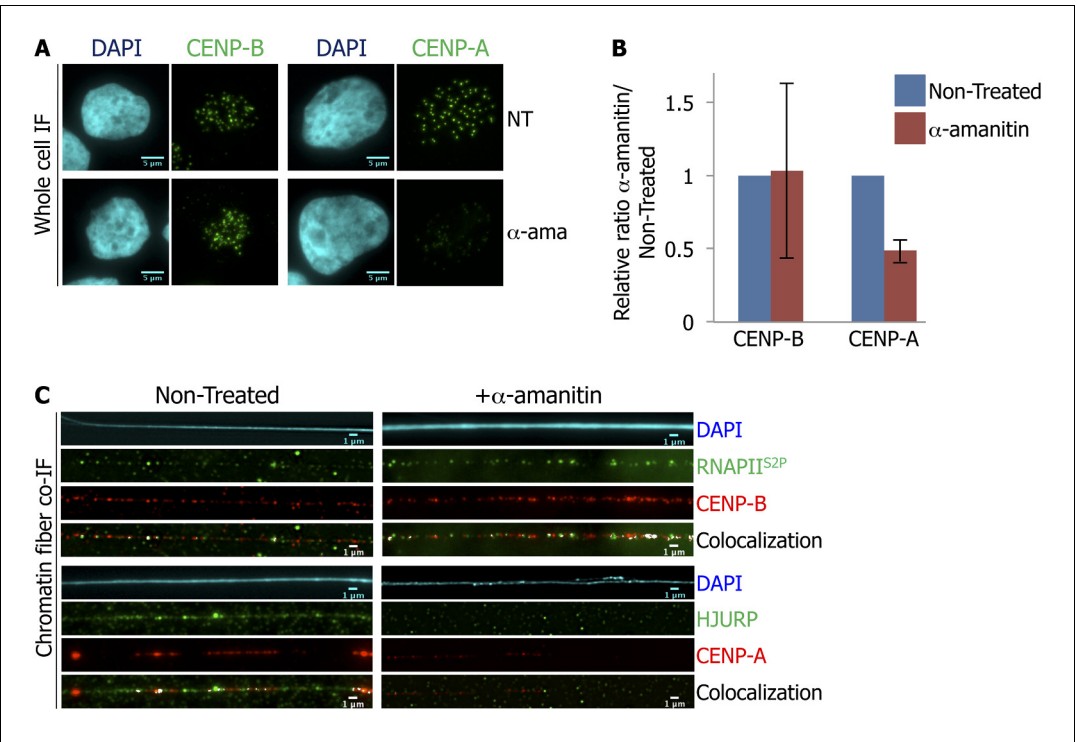

**Figure 3.** RNA Polymerase II (RNAPII)-dependent transcription is required for the recruitment of CENP-A and its chaperone HJURP onto the centromere at early G1 (eG1). (**A**) eG1 synchronized HeLa cells were treated or not (NT) with α-amanitin (α-ama) for 2 hr before staining for centromeric proteins CENP-A or CENP-B (green). The DAPI raw image is shown for a representative cell (cyan). Three independent experiments were performed and in each, a minimum of 30 cells were analyzed per slide. Scale bar: 5 μm. (**B**) Signal intensity of CENP-A and CENP-B spots from (A) was quantified using ImageJ, and relative ratios of α-amanitin vs non-treated conditions were determined. Means ± SD from three independent experiments is represented on the graph. Quantification values are tabulated in *Supplementary file 1*. (**C**) Centromeric proteins CENP-A and CENP-B (red) were co-stained with HJURP (green) and RNAPII phosphorylated on serine 2 (RNAPII$^{S2P}$, green), respectively, on centromeric chromatin fibers prepared from eG1 synchronized cells treated or not with α-amanitin for 2 hr (cyan, DAPI). Three independent experiments were performed and in each, a minimum of five chromatin fibers were analyzed per slide (co-localization on the same chromatin fiber after α-amanitin compared to non-treated: between CENP-B and RNAPII$^{S2P}$ = 9/15 vs 10/15; and between CENP-A and HJURP = 2/15 vs 9/15). Scale bar: 1 μm. Co-IF, co-immunofluorescence.

DOI: https://doi.org/10.7554/eLife.03254.007

The following figure supplement is available for figure 3:

**Figure supplement 1.** Inhibition of RNA Polymerase II-dependent transcription at early G1 (eG1) does not affect CENP-A and HJURP protein levels.

DOI: https://doi.org/10.7554/eLife.03254.008

RNAPII was blocked, the intensity of the CENP-A signal was decreased by ~50% in cells (*Figure 3A, B*; *Supplementary file 1*). To ensure that this decrease was not due to reduced expression of CENP-A or its chaperone HJURP, total levels of both proteins were quantified by Western blot. No noticeable changes in protein levels were detected upon α-amanitin treatment (*Figure 3—figure supplement 1B*). These data indicate that equal amounts of CENP-A and HJURP were available at eG1, but potentially unable to load at the centromere.

The decrease of CENP-A signal at centromeres during eG1 after RNAPII inhibition might be due to either the loss of pre-existing CENP-A, or a defect in targeting of newly synthesized CENP-A by its chaperone HJURP. To discriminate between these two hypotheses, we analyzed the localization of CENP-A, CENP-B, RNAPII$^{S2P}$, and HJURP on chromatin fibers, with or without α-amanitin treatment at eG1. Consistent with the known effect of α-amanitin on RNAPII (i.e., blocking RNAPII elongation without inducing the release of RNAPII) (*Nguyen et al., 1996*; *Bensaude, 2011*), inhibition of

transcription did not affect RNAPII$^{S2P}$ localization onto the centromeric fiber (*Figure 3C*, first top panel). Similarly, CENP-B localization onto centromeric chromatin fibers remained unaffected as well (*Figure 3C*, first panel). Consistent with the whole cell data presented above (*Figure 3A,B*), after RNAPII inhibition, not only were CENP-A signals reduced on chromatin fibers (*Figure 3C*, second panel), HJURP was almost completely lost (*Figure 3C*, second panel). The loss of HJURP (*Figure 3C*) combined with the ~50% decrease of CENP-A signal after α-amanitin treatment (*Figure 3B*), suggests that RNAPII-dependent transcription is required for the targeting of HJURP, and for the subsequent loading of new CENP-A to the centromeric chromatin fiber at eG1.

## A 1.3 kb centromeric RNA binds the soluble HJURP/CENP-A pre-assembly complex at early G1 (eG1)

Previous data have documented the existence of ncRNA at centromeres in multiple species (*Hall et al., 2012*). In humans, no genes have been annotated within native centromeres, suggesting a transcription event at α-satellite DNA repeats most likely leads to the synthesis of ncRNAs. In order to characterize potential centromeric transcripts, we sought to purify them biochemically. Total RNAs were extracted from cells, DNase I treated to remove genomic contamination, separated on denaturing gels, transferred to Northern blots, and subjected to hybridization with radiolabelled centromeric α-satellite probes, in order to reveal potential complementary transcripts. Northern blots revealed a unique centromeric RNA species migrating at approximately ~1.3 kb (*Figure 4A*, *Figure 4—figure supplement 1A*). Control experiments were performed to exclude the possibility of trace genomic DNA contamination contributing to the 1.3 kb band. Treatment of RNA samples with RNase A (*Figure 4—figure supplement 1B*), or purification of RNA from cells treated with α-amanitin (*Figure 4—figure supplement 1C*), both demonstrated the absence of the 1.3 kb band on Northern blots, supporting the interpretation that the 1.3 kb band derives solely from an RNA species.

The inhibition of transcription was accompanied by the loss of CENP-A and HJURP at the centromere during eG1 (*Figure 3C*), and our results above (*Figure 4A*) supported the possibility of a unique RNA species present at centromeres in eG1. A logical prediction arising from these data is that centromeric transcripts might physically associate with the soluble pre-assembly HJURP/CENP-A complex in vivo. Indeed, computational RNA-binding prediction algorithms revealed potential RNA binding residues in both HJURP and CENP-A (*Figure 4—figure supplement 2*; *Wang et al., 2010*). Thus, to further test this hypothesis, we probed for physical interactions between CENP-A and its chaperone HJURP with centromeric α-satellite transcripts.

After a brief MNase digestion of eG1-synchronized cells, we immunoprecipitated CENP-A and HJURP from both, the soluble fraction (composed of free histones and nuclear factors, SF), and the chromatin fraction (composed of chromatin and associated complexes, CF) (Experimental Scheme, *Figure 4—figure supplement 3*). CENP-A and HJURP complexes were immunoprecipitated from SF and CF. Mock IPs pulled down neither CENP-A nor HJURP (*Figure 4—figure supplement 4A*). Consistent with HJURP chaperoning CENP-A at eG1 (*Dunleavy et al., 2009*; *Foltz et al., 2009*; *Shuaib et al., 2010*), these proteins co-purified from both SF and CF (*Figure 4—figure supplement 4A*). From these IPs, RNAs were purified, electrophoresed, transferred to Northern blots, and subsequently hybridized to the same radiolabelled centromeric α-satellite probes as above (*Figure 4A*). These Northern blots revealed no RNA signal in the mock IP (*Figure 4—figure supplement 4B*). In contrast, the 1.3 kb RNA is physically associated with CENP-A in both SF and CF (*Figure 4B*), and interacts with HJURP only in the SF (*Figure 4B*). These data provide evidence that the 1.3 kb centromeric RNA physically associates with the soluble HJURP/CENP-A pre-assembly complex at eG1.

## CENP-A associated RNA localizes to centromeres

We next sought to purify, clone using a conventional TOPO T/A cloning strategy and sequence CENP-A-associated RNA (Experimental Scheme, *Figure 4—figure supplement 3*). This sequencing approach was moderately successful, yielding one sequence of ~675 nucleotides (cenRNA#1, *Figure 5—figure supplement 1*). This RNA sequence is unique and contains four semi-regular spaced 28 bp repeats with a weak homology (~52%) to the canonical CENP-B box (*Supplementary file 2*), but does not map to the currently annotated human genome sequence, to any other organisms, or to plasmids. Over the course of the subsequent two years after publication, we made additional

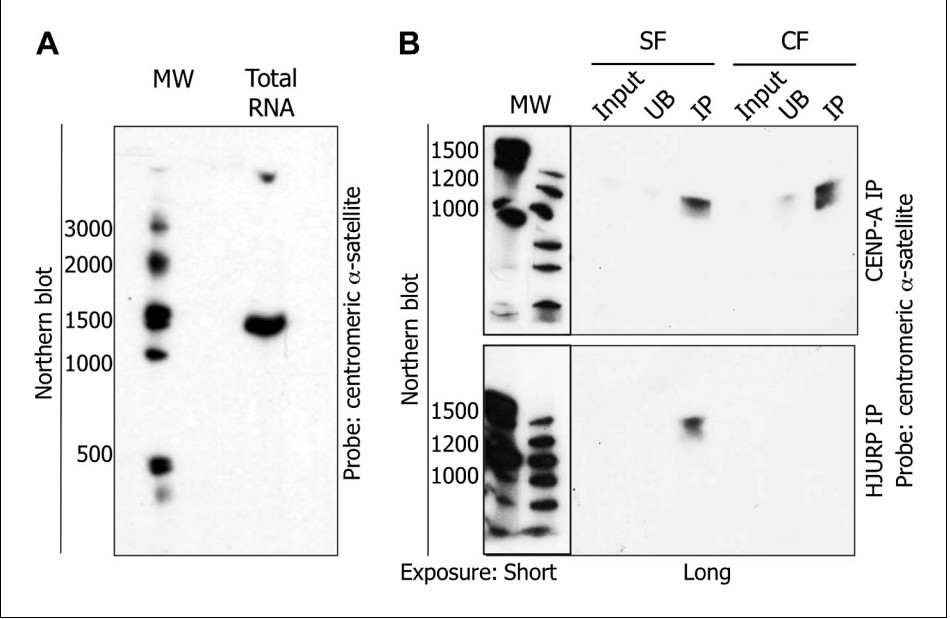

**Figure 4.** The pre-assembly soluble HJURP/CENP-A complex binds a 1.3 kb centromeric transcript at early G1 (eG1). (**A**) Total RNAs from HeLa cells were separated on denaturing gel and visualized by Northern blot with radiolabeled centromeric α-satellite probes. (**B**) Co-immunoprecipitated RNAs by CENP-A or HJURP were analyzed by Northern blot as in (**A**). CF, chromatin fraction; IP, immunoprecipitate; MW, molecular weight; SF, soluble fraction; UB, unbound.

DOI: https://doi.org/10.7554/eLife.03254.009

The following figure supplements are available for figure 4:

**Figure supplement 1.** Centromeric transcripts are 1.3 kb in length.

DOI: https://doi.org/10.7554/eLife.03254.010

**Figure supplement 2.** HJURP and CENP-A display potential RNA binding residues.

DOI: https://doi.org/10.7554/eLife.03254.011

**Figure supplement 3.** Scheme presenting the strategy for RNA-chromatin immunoprecipitation, RNA purification, and sub-cloning for sequencing.

DOI: https://doi.org/10.7554/eLife.03254.012

**Figure supplement 4.** Mock IPs demonstrate specificity of the 1.3 kb RNA binding to CENP-A and HJURP.

DOI: https://doi.org/10.7554/eLife.03254.013

attempts to map centromeric RNAs, turning to a high-throughput approach coupled to CENP-A and HJURP RIP-Seq. These data yielded ~435 centromeric-mapping RNA sequences (*Quénet et al., 2016*), which, by RNA FISH localize to centromeres (unpublished). However, the initial cloned Cen-RNA#1 sequence did not map to this database of centromeric RNAs. This led us to re-sequence 6 clones of CenRNA#1, which led to the rescue of 12 N's and 27 base calls changed in the 675bp sequence (*Figure 1*). We then performed an expanded sequence search across multiple sequence databases beyond the publicly available NCBI catalog, including the HeLa genome (after obtaining permission from the HeLa DGAP working group), as well as an industrial database for cloning vectors *Supplementary files 2*, *3*, *4*. Much to our dismay, we discovered that cenRNA#1 arose largely from fusions of adapters used in the RNA cloning approach and described in Pfeffer et al. (*Figure 2*; *Pfeffer et al., 2005*). Based on this evidence, we no longer believe that the original sequence provided for putative cenRNA#1 represents a human centromeric transcript.

In our initial version, we seek to assess the functional role of cenRNA#1 by an shRNA strategy to down-regulate specifically its expression (*Figure 5—figure supplement 1*). The two shRNAs were generated from the putative sequence of cenRNA#1, against the 28bp repeat element sequence. Cells were transfected with control scrambled (shRNA^scram) or shRNA^cenRNA#1 constructs, and selected with puromycin Cells transfected with shRNA^scram displayed no changes in morphology and density, whereas cells treated with shRNA^cenRNA#1 displayed significant loss of cell density (down by

~70%, relative to control), and presented aberrant morphology (*Figure 5—figure supplement 4A and 2B*).

We decided to readdress this experiment. Based on the re-sequencing results, three base calls within each shRNA were changed, which removes their uniqueness within cenRNA#1 and changes the percent identity to 26/29 bases (*Supplementary file 6*). Neither shRNA had a significant match to the human genome or known transcripts. However, the best hit is to a lncRNA on chromosome 3 (15/29 bases), which is proximal to the centromere, but not in the pericentromere (*Supplementary file 7*). This proximity may potentially explain the positive IF/FISH signal observed in our initial manuscript of CENP-A with cenRNA#1DNA probe (*Figure 5*). An independent reproduction of the down-regulation of cenRNA#1 by shRNA approach yielded the same chromosome defect as before (M. Bui, data not shown). To re-test whether some fraction of cenRNA#1 matching sequence plays a role in chromosomal integrity, we next designed a locked nucleic acid antisense oligonucleotide (LNA ASO) and analyzed mitosis integrity. Indeed, LNA ASO targeting cenRNA#1 led to modestly increased rates of lagging chromosomes and cells with multi-polar spindles, when compared to either un-transfected, mock transfected, or scrambled transfected cells (28% and 13% compared to 10% and 3% for scrambled control; (*Figure 5—figure supplement 5*). This result may suggest a potential function for some fraction of this non-centromeric sequence on chromosome segregation and mitotic integrity, but which is not connected to the main findings of our original manuscript.

## Targeting destruction of centromeric α-satellite transcripts results in severe mitotic defects

We were curious whether it was the act of centromeric transcription alone, or the product of transcription (i.e., centromeric RNAs), that was necessary for HJURP and CENP-A targeting to the centromere at eG1. To distinguish between these hypotheses, we further examined functional consequences arising from the targeted loss of total centromeric α-satellite RNA, without inhibiting RNAPII transcription.

Using the centromeric α-satellite consensus sequence (*Waye and Willard, 1987*), we designed two shRNAs targeting α-satellite sequences (shRNA$^{sat1}$ and shRNA$^{sat2}$) to destroy centromeric transcripts (*Figure 6—figure supplement 1A*).

At 6 days post-transfection with the control scrambled (shRNA$^{scram}$) or shRNA$^{sat}$ constructs and puromycin selection, the expression level of centromeric α-satellite transcript was analyzed by qtPCR. Compared to control shRNA$^{scram}$, cells transfected with shRNA$^{sat}$ constructs displayed a significant decrease (~70%) of the centromeric α-satellite transcript (*Figure 6— figure supplement 1B*), confirming targeted destruction was accomplished.

Evaluation of cell morphology by phase contrast microscopy revealed that shRNA$^{sat}$-transfected cells were less dense (down by ~70%, relative to control) and exhibited abnormal morphology (*Figure 6A*). Phenotypes included a large and flat cytoplasm, cellular protrusions, and multinucleate cells (*Figure 6A*). To better elucidate cell defects, we stained with β-actin, which revealed cells with several nuclei and atypical shape (*Figure 6B*). Reduced cell density and morphological abnormalities could result from cells exiting the replicative cell cycle and undergoing senescence (*Kuilman et al., 2010*). We performed β-galactosidase staining to test for senescence (*Bandyopadhyay et al., 2005*). Senescent BJ cells were used as positive control, and displayed the expected blue color after the β-galactosidase assay (*Figure 6—figure supplement 2*). However, no significant increase in senescence was seen in either shRNA$^{scram}$ or shRNA$^{sat}$-transfected cells (*Figure 6—figure supplement 2*).

The kinds of cellular morphological changes observed above (*Figure 6A,B*) have previously been linked to defects in cell division, specifically in mitosis (*Carone et al., 2013*). To test this alternative possibility, shRNA-transfected cells were synchronized at mitosis, and at 6 days post-transfection, stained for markers of mitotic spindles (α-tubulin) and centromeres (CENP-B), respectively. shRNA$^{scram}$-transfected cells displayed normal mitotic structures (*Figure 6C*). In contrast, almost half (42.2%) of shRNA$^{sat}$-transfected cells presented abnormal mitoses, with multipolar spindles and lagging chromosomes (*Figure 6C*).

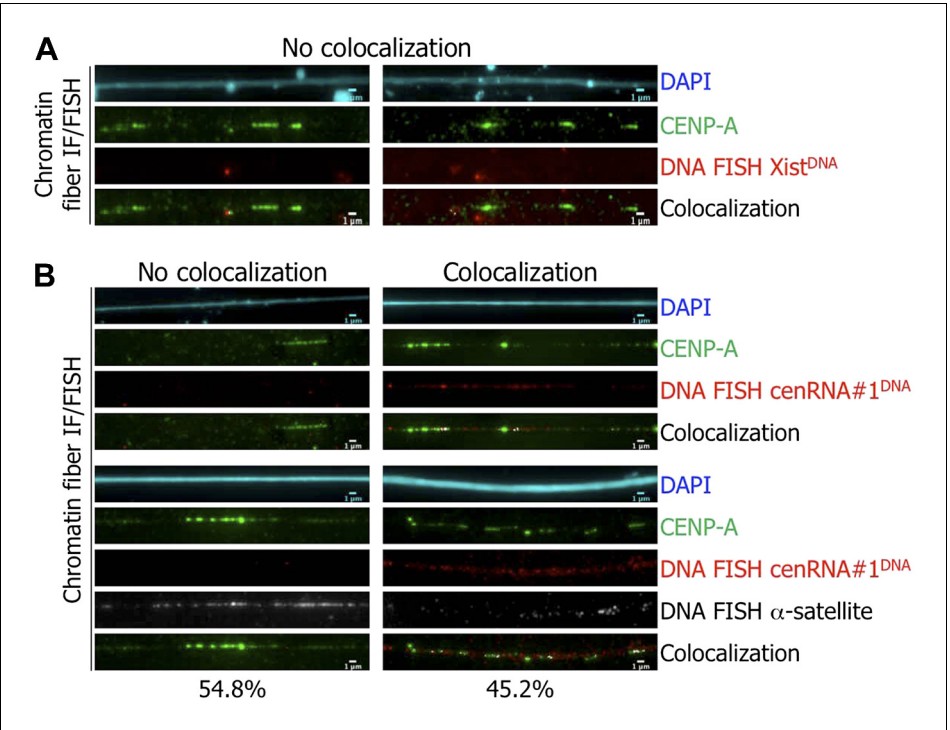

**Figure 5.** The identified cenRNA#1 transcript associated with CENP-A has a centromeric origin. (**A**) CENP-A (green) was co-stained with Xist DNA FISH probe (FISH Xist[DNA], red) on chromatin fibers (cyan, DAPI). Two independent experiments were performed and in each, a minimum of eight chromatin fibers were analyzed per slide (co-localization on the same chromatin fiber between CENP-A and FISHXist[DNA] = 0/18). Scale bar: 1 μ m. (**B**) Chromatin fibers were either co-stained by IF/FISH with CENP-A (green) and cenRNA#1[DNA] (red), or IF/double-FISH with CENP-A (green), cenRNA#1[DNA] (red) and centromeric α-satellite probes (gray). The DAPI raw image is shown for a representative chromatin fiber (cyan). Two independent experiments were performed and in each, a minimum of eight chromatin fibers were analyzed per slide (co-localization on the same chromatin fiber between CENP-A and cenRNA#1[DNA] and FISH α-satellite compared to no co-localization = 9/20 vs 11/20). Scale bar: 1 μm. FISH, fluorescencein situ hybridization; IF, immunofluorescence.

DOI: https://doi.org/10.7554/eLife.03254.014

The following figure supplements are available for figure 5:

**Figure supplement 1.** Sequence of cenRNA#1.
DOI: https://doi.org/10.7554/eLife.03254.015
**Figure supplement 2.** Alignment of the original cenRNA1 with new sequencing results.
DOI: https://doi.org/10.7554/eLife.03254.025
**Figure supplement 3.** Map of re-sequenced cenRNA#1
DOI: https://doi.org/10.7554/eLife.03254.026
**Figure supplement 4.** shRNA[cenRNA#1]-transfected cells have a cell survival defect.
DOI: https://doi.org/10.7554/eLife.03254.016
**Figure supplement 5.** The down-regulation of cenRNA#1 leads to chromosome defects. shRNA[cenRNA#1]-transfected cells have a cell survival defect.
DOI: https://doi.org/10.7554/eLife.03254.027

## Targeting destruction of centromeric α-satellite transcripts results in abrogation of CENP-A and HJURP targeting at early G1 (eG1)

A mechanistic explanation for the observed mitotic aberrances in the centromeric transcript depleted cells (*Figure 6C*) could be loss of centromere integrity, driven by deficient targeting of HJURP/CENP-A complexes to centromeres. We were curious whether the loss of centromeric transcripts directly abrogated CENP-A and HJURP localization at centromere. Because the loss of centromeric transcripts resulted in reduced cell density and mitotic defects (*Figure 6*), there were insufficient cells for biochemical experiments. Thus, we turned to chromatin fibers to investigate this

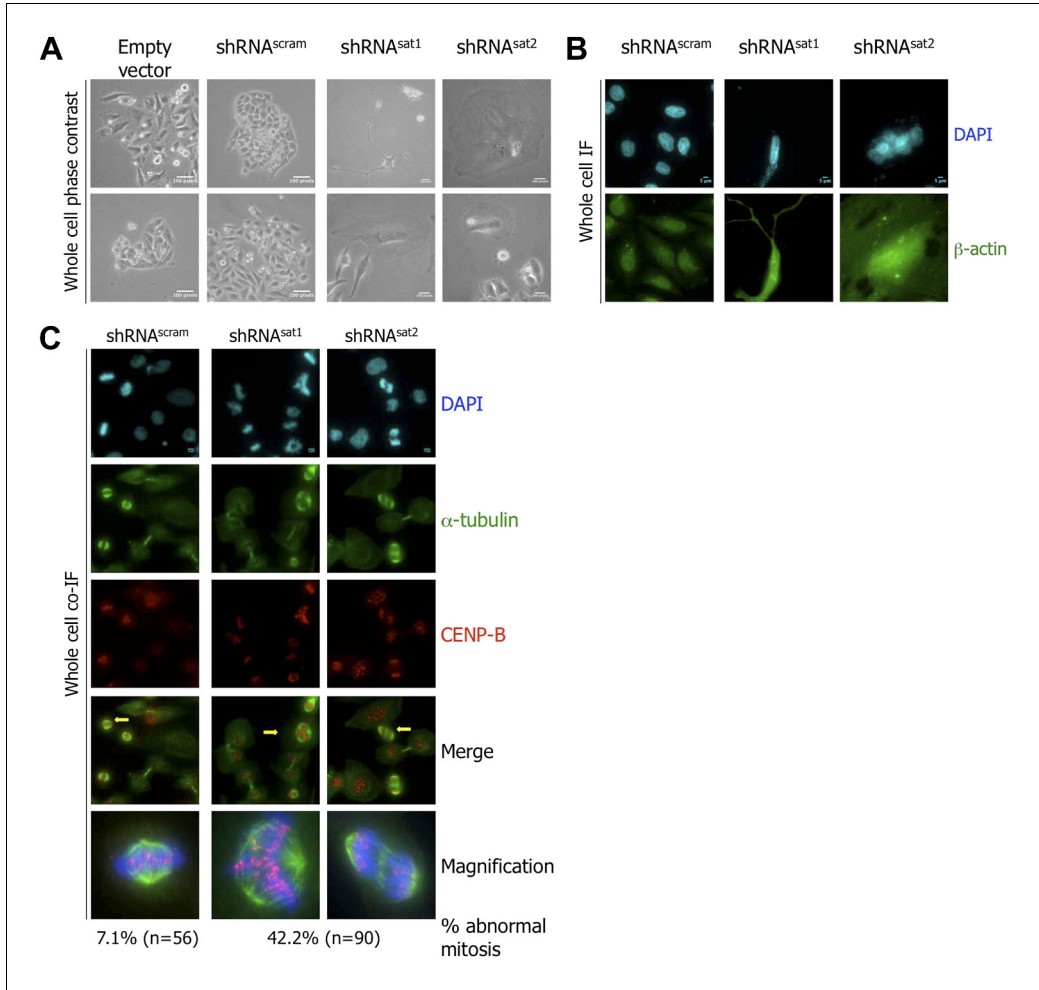

**Figure 6.** Targeted down-regulation of centromeric α-satellite transcript results in mitotic defect. (**A**) Six days post-transfection of empty vector, control scrambled (shRNA$^{scram}$), or α-satellite shRNA (shRNA$^{sat1}$, shRNA$^{sat2}$), cell morphology was observed by phase contrast microscopy. Scale bar: 100 pixels. (**B**) Cells were treated as in (A) and stained for β-actin (green) and DAPI (cyan) to reveal alterations in cell morphology. Scale bar: 5 μm. (**C**) Cells were treated as in (A), synchronized in mitosis and stained for α-tubulin (green), CENP-B (red) and DAPI (cyan/blue). Three independent experiments were performed and in each, a minimum of 30 cells were analyzed per slide. Scale bar: 5 μm. IF: immunofluorescence.

DOI: https://doi.org/10.7554/eLife.03254.017

The following figure supplements are available for figure 6:

**Figure supplement 1.** Down-regulation of centromeric RNAs by a targeted shRNA approach.
DOI: https://doi.org/10.7554/eLife.03254.018

**Figure supplement 2.** Down-regulation of centromeric transcripts does not induce senescence.
DOI: https://doi.org/10.7554/eLife.03254.019

issue. After synchronization at eG1, chromatin fibers were isolated from shRNA$^{scram}$ cells, or the few remaining of shRNA$^{sat}$-transfected cells, and stained for RNAPII and centromeric proteins. In both, shRNA$^{scram}$- or shRNA$^{sat}$-transfected cells, RNAPII$^{S2P}$ remains associated with centromeric chromatin fibers (*Figure 7*, first top panel), confirming that RNAPII localization and transcription of centromeres are independent of centromeric transcripts. Additionally, CENP-B and CENP-C localization was also unaffected at centromeres (*Figure 7*, second and third panels). In contrast, on centromeric fibers from shRNA$^{sat}$-transfected cells, CENP-A and HJURP were barely detectable (*Figure 7*, fourth and fifth panels). These data suggest that at eG1, CENP-A targeting through its chaperone HJURP is dependent not just on active transcription itself, nor on processes that facilitate centromeric transcription (*Barnhart et al., 2011*; *Wang et al., 2014*), but specifically requires the presence of

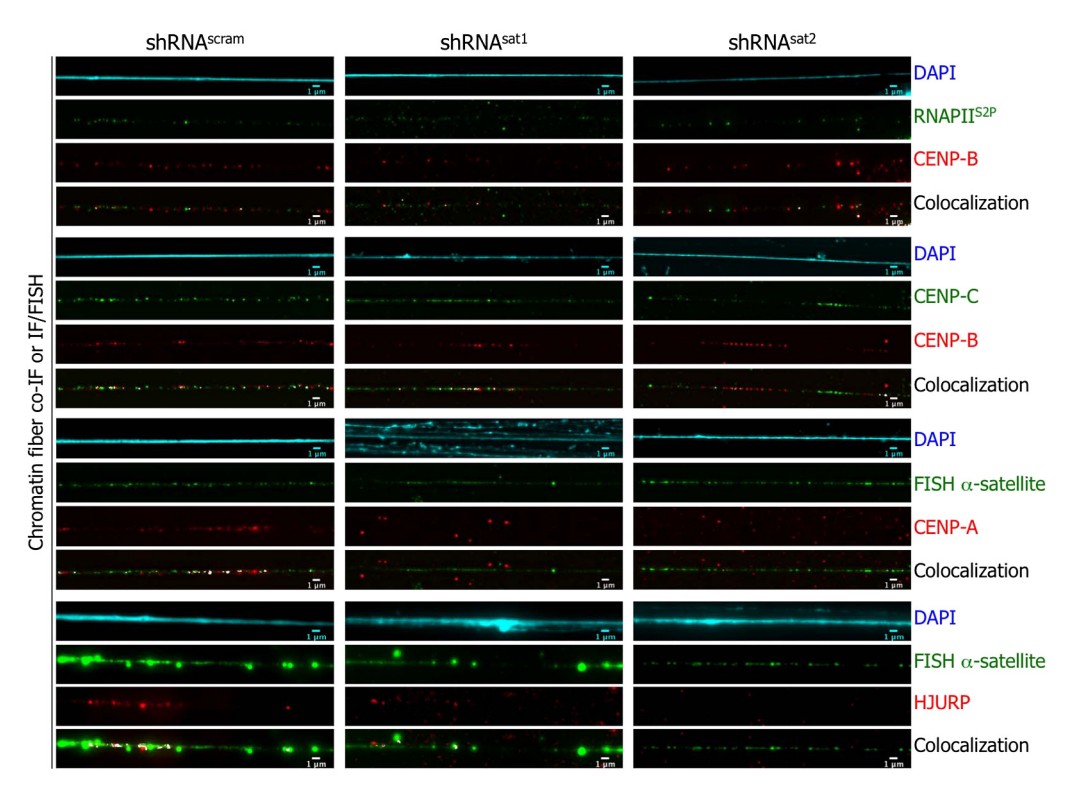

**Figure 7.** Targeted down-regulation of centromeric α-satellite abrogates HJURP/CENP-A targeting to centromeric chromatin at early G1 (eG1). Chromatin fibers were prepared from shRNA[scram], shRNA[sat1], or shRNA[sat2]-transfected cells synchronized at eG1. To visualize centromeric domains, co-IF was performed for CENP-B (red), CENP-C (green) and RNA Polymerase II (phosphorylated on serine 2, RNAPII[S2P], green), whereas CENP-A (red) and HJURP (red) antibodies were co-stained with a DNA FISH probe against centromeric α-satellite DNA repeats (green). The DAPI raw image is shown for a representative chromatin fiber (cyan). Three independent experiments were performed and in each, five chromatin fibers were analyzed per slide (co-localization on the same chromatin fiber in shRNA[sat1] or shRNA[sat2]-transfected cells: between CENP-B and RNAPII[S2P] = 9/15 and 8/15; CENP-B and CENP-C = 14/15 and 13/15; between CENP-A and centromeric α-satellite DNA = 3/15 and 1/15; between HJURP and centromeric α-satellite DNA = 1/15 and 0/15, respectively). Scale bar: 1 μm. FISH, fluorescence in situ hybridization; IF, immunofluorescence.
DOI: https://doi.org/10.7554/eLife.03254.020

The following figure supplement is available for figure 7:

**Figure supplement 1.** .A speculative model proposing a mechanism by which centromeric long non-coding RNAs target soluble HJURP/CENP-A complexes to centromeres at early G1 (eG1).
DOI: https://doi.org/10.7554/eLife.03254.021

centromeric transcripts. The long-term loss of these centromeric transcripts leads to mitotic defects (*Figure 6*), which are physically underpinned by the loss of HJURP recruitment and CENP-A loading (*Figure 7*) to native centromeres.

## Discussion

Active transcription is thought to be essential for centromere structure and function (*Saffery et al., 2003*; *Wong et al., 2007*; *Bergmann et al., 2011*; *Chan et al., 2012*). In this study, we investigated the mechanistic contribution of transcription, and centromeric transcripts, to centromeric integrity. We show that RNAPII and TBP are loaded onto and transcribe human centromeric chromatin at eG1 (*Figure 1*). This cell cycle regulated centromeric transcription is required for the synthesis of centromeric RNAs (*Figure 2*). Biochemical purification and analysis reveal a 1.3 kb transcript which is physically associated with CENP-A and HJURP in the soluble pre-assembly state (*Figure 4*). Targeted destruction of this centromeric RNA leads to the loss of centromere integrity and subsequent mitotic and cellular defects (*Figure 6*), which are mechanistically underpinned by the loss of HJURP and

CENP-A recruitment to centromeres at eG1 (*Figure 7*). Altogether, these data reveal a hitherto unsuspected function for lncRNAs in RNA-dependent chaperone targeting to centromeres in human cells (*Figure 7—figure supplement 1*).

Several specific questions arise from our observations. First, no active genes have ever been described in human centromeres, making the identification of RNAPII (*Figure 1*; *Saffery et al., 2003*) and TBP at the centromere surprising. Because the repetitive nature of centromeric α-satellite DNA has thus far disallowed complete sequencing, successful annotation of transcriptional motifs that may exist in human centromeres remains to be accomplished. Our data show that centromeric transcription is an event integral to the epigenetic maintenance of centromere integrity, and discovering precisely where such motifs lie within active centromeres is an exciting avenue of research.

Second, our functional characterization of a 1.3 kb centromeric lncRNA deriving from centromeric transcription reveals its interaction with CENP-A and HJURP at eG1. Specific depletion of centromeric α-satellite transcripts affects the recruitment of both CENP-A and HJURP proteins, directly implicating centromeric RNA in CENP-A and HJURP targeting onto centromeres at eG1. We note that previous data have shown that HJURP loading is also dependent on its interaction with the Mis18 complex, in a CDK1-dependent manner (*Barnhart et al., 2011*; *Moree et al., 2011*; *Müller et al., 2014*; *Wang et al., 2014*). However, mutations of CDK1-phosphorylated sites in HJURP only partially abrogate its recruitment in vivo, highlighting the existence of other hitherto unknown CENP-A loading factors (*Wang et al., 2014*). We speculate centromeric lncRNAs are, in fact, the missing factor. Thus, of immediate interest is the elucidation of the structure of the 1.3 kb centromeric RNA with its cognate binding domains in CENP-A and HJURP.

Third, the exact molecular process involved in targeting lncRNA-nucleoprotein complexes to centromeres is an unexpected and novel avenue to pursue. For example, whereas it is well known that Xist RNA binds its cognate DNA locus only in cis (*Plath et al., 2002*), it is unknown if centromeric transcripts can bind in cis solely to the centromere of origin, or in trans, across all centromeres. An attractive possibility is centromeric RNAs originate from multiple centromeres and serve a redundant function to ensure accurate targeting of CENP-A/HJURP to homologous centromeres.

Fourth, our study has potential evolutionary implications. Prior studies have described RNA originating from centromeres in multiple species (*Bouzinba-Segard et al., 2006*; *Carone et al., 2009*). In mouse cells, a 120 nucleotide minor satellite RNA is associated with centromeres (*Bouzinba-Segard et al., 2006*), and in tammar wallaby, centromeric transcription results in the production of ~40 nucleotides crasiRNAs (centromere repeats-associated short interacting RNAs) (*Carone et al., 2009*; *Lindsay et al., 2012*; *Carone et al., 2013*). A logical explanation for the difference in size of ncRNAs generated in different organisms may be the divergent nature of the centromeric DNA sequences across species, which in turn may lead to divergence in the type of centromeric RNAs produced. However, despite this difference, over-expression or down-regulation of mouse minor satellite RNA, or crasiRNAs in tammar wallaby, or the 1.3 kb centromeric human RNA identified in our study, leads to similar cellular and mitotic defects. Our data reveal that such RNAs generated from human centromeric transcription bind HJURP and CENP-A in the soluble form and that mitotic loss seen in cells depleted of these lncRNAs is specifically linked to abrogation of HJURP-mediated targeting of CENP-A. Thus, our data suggest an evolutionarily conserved basis for the phenomena of centromeric transcription seen in other organisms. We speculate that accurate CENP-A targeting onto active centromeres probably requires a dual-lock system, coupling chromatin-bound centromeric factors (such as Mis18), which facilitate cell-cycle regulated centromeric transcription, which in turn results in the production of a lncRNA/CENP-A/chaperone complex that can effectively target CENP-A back to pre-existing active centromeric sites (*Figure 7—figure supplement 1*).

It is noteworthy that transcription-coupled, chaperone-mediated histone variant assembly governs much of chromatin biology. Our report potentially reveals an RNA-based mechanism by which specialized histone-variant driven chromatin structures might be maintained in vivo.

## Materials and methods

### LNA ASO sequence and LNA probes
LNA ASOs were designed and purchased by QIAGEN (previously Exiqon). *Supplementary file 9* lists sequences of these LNA ASO sequence.

## Down-regulation of cenRNA#1

Transfection of HeLa cells with LNA ASO were conducted as in (Bui et al., 2017). Briefly, cells were seeded 24 hours before transfection to allow no more than 75% confluency, and transfected using Lonza's Amaxa Cell Line Nucleofector Kit R (Cat #VCA-1001) with Amaxa Biosystems Nucleofector II electroporation system using program O-005. After transfection, cells were grown on coverslips with fresh DMEM medium.

## Immuno-fluorescence of mitotic cells

The day following LNA ASO transfection, cells were synchronized with a double thymidine block and released for 10.5 hours on the third day, to enrich for anaphase-cytokinesis staged cells. IF experiments were performed as described previously (*Bui et al., 2012*). Cells were fixed with 4% paraformaldehyde in 1X PBS (#14190-144; Gibco by Life Technologies) for 15 min, permeabilized with 0.5% Triton X-100 in 1X PBS for 10 min, and blocked with 3% bovine serum albumin (BSA, #BP9706-100; Fisher Scientific) in 1X PBS. Coverslips were immuno-stained for CENP-C and α-tubulin for one hour each (*Supplementary file 8*). After three washes in 1X PBS, 0.1% Tween (#P7949-500ML; Sigma-Aldrich), cells were incubated with secondary antibody (goat anti-guinea pig or anti-mouse IgG (H +L) secondary antibodies, Alexa Fluor568 and Alexa Fluor488 conjugates (Thermo Fisher Scientific)) in 1X PBS for 1 hour at RT in the dark. Finally, cells were washed three times for 5 min at RT. Coverslips were mounted with anti-fade mounting medium Prolong Gold with DAPI.

## Microscopy observation and analysis

IF slides were observed with a DeltaVision Elite RT microscopy imaging system (GE Healthcare) controlling an interline charge-coupled device camera (Coolsnap) mounted on an inverted microscope (IX-70; Olympus). Images were captured by using a 60X objective at 0.2μm z-sections and analyzed with Image J (1.50e; Java 1.6.0_20).

## Antibodies

Antibodies are commercially available, except the custom CENP-A antibody (available upon request) used for CENP-A detection on Western blot. *Supplementary file 8* lists all antibodies used for each experiment.

## Cell culture and RNA polymerase inhibition

HeLa cells were grown at 37°C in a humidified atmosphere containing 5% $CO_2$, in Dulbecco's modified Eagle's medium high in glucose and L-glutamine (#11965; Gibco, Grand Island, NY) supplemented with 10% Fetal Bovine Serum (#26140 – 079; Gibco) and 1X Pen/Strep solution (#10378 – 016; Gibco).

All synchronizations were done by double thymidine block (0.5 mM, #T9250; Sigma-Aldrich, Saint Louis, MO). After a first block of 19 hr, cells were released for 9 hr, followed by a second thymidine block of 16 hr. Cells then were released for the appropriate time (9 hr for G2, 10 hr for mitosis, and 11 hr for eG1, *Figure 1—figure supplement 1*). Synchronization was assessed by flow cytometry. Cells were stained with propidium iodide (#P817045, Invitrogen, Grand Island, NY) and analyzed on a FACScalibur (Becton Dickinson, San Jose, CA). Synchronized cells were treated with either 0.2 μg/ml of actinomycin D (#A2263, Sigma-Aldrich) or 2 μg/ml of α-amanitin (#A1410; Sigma-Aldrich) to analyze the effect of RNAPI and RNAPII inhibition, respectively, on centromere transcription.

## RNA extraction, retro-transcription and polymerase chain reaction (PCR)

RNAs were extracted by Trizol reagent (#15596 – 026; Ambion, Grand Island, NY) according to manufacturer protocol. Briefly, cells were resuspended in Trizol, and following 5 min incubation at room temperature (RT), 200 μl of chloroform (#BP1145-1, Fisher Scientific, Pittsburgh, PA) was added. After centrifugation at 12,000 rpm for 15 min at 4°C, the clear phase was mixed with 500 μl of isopropanol (#534021, Sigma-Aldrich) and centrifuged. The pellet was washed with 75% ethanol (diluted from 100% ethanol, #61509 – 0010, Acros Organics, Pittsburgh, PA) and resuspended in water complemented with DNase I buffer, DNase I (#M0303; New England Biolabs NEB, Ipswich, MA), and RNase inhibitor (#M0314, NEB) to avoid genomic DNA contamination. After incubation for

30 min at 37°C, the DNase I activity was inhibited by addition of 5 mM EDTA (#351-027-721, Quality Biological, Gaithersburg, MD) and incubation at 65°C for 10 min. RNAs were purified a second time by phenol:chloroform:isoamyl alcohol (25:24:1; #AC327115000) method and ethanol precipitated. RNAs were conserved at −80°C until further analysis.

After quantification by UV-spectroscopy (230, 260, and 280 nm) and verification of RNA quality on 1.5% agarose gel, equivalent concentrations of RNA were subjected to retro-transcription, using the SuperScript III First-Strand Synthesis System with random hexamer primers (#18080 – 051), and amplified with Takara PCR kit (#RR001B; Clontech Laboratories Inc., Mountain View, CA). Control reactions without the reverse transcriptase or complementary DNA were performed to rule out DNA contamination and non-specific amplification, respectively. Primer sequences are included in *Supplementary file 3* (*Dunham et al., 1992*). PCR conditions were defined for each analyzed sequence. The setup for GAPDH and centromeric α-satellite were 3 min 94°C; [10 s 98°C, 30 s 57°C, 30 s 72°C] 30 cycles; 5 min 72°C. The conditions for 18S rRNA were 3 min 94°C; [10 s 98°C, 30 s 52°C, 30 s 72°C] 30 cycles; 5 min 72°C. Finally, the PCR status for cenRNA#1 were 3 min 94°C; [10 s 98°C, 30 s 57°C, 30 s 72°C] 45 cycles; 5 min 72°C.

## Immunofluorescence (IF)

Cells were grown on poly-D-Lysine-treated coverslips in six-well plate and synchronized by double thymidine block. After two washes with cold 1× PBS, they were prefixed for 30 s with cold 4% para-formaldehyde (PFA, #15714 s; Electron Microscopy Sciences, Hatfield, PA) in PEM (80 mM K-PIPES pH6.8, 5 mM EGTA pH7.0, 2 mM MgCl$_2$). Following three washes with cold PEM, soluble proteins were extracted for 5 min on ice with 0.5% Triton X-100 (#327372500, Acros Organics) in CSK (10 mM PIPES pH6.8, 100 mM NaCl, 300 mM sucrose, 1 mM EGTA, 3 mM MgCl$_2$). Few drops of 4% PFA in PEM were added for 5 min. Slides were then incubated with fresh 4% PFA in PEM for 40 min on ice. After three washes with PEM, cells were permeabilized with 0.5% Triton X-100 in PEM for 30 min at RT, washed again three times, and blocked in 1×— TBS, 3% Bovine Serum Albumin, 5% normal goat serum (#005-000-121; Jackson ImmunoResearch, West Grove, PA) for 1 hr at RT. Finally cells were incubated with the primary antibody diluted in 1×— TBS, 1% Bovine Serum Albumin, 5% normal goat serum over-night (O/N) at 4°C in a humidified chamber. Slides were washed three times for 5 min at RT with 1×— TBS, 0.1% Tween 20 (#P7949, Sigma-Aldrich), and incubated with secondary antibody for 1 hr at RT. After washing, the same protocol was repeated for co-IF, and then cells were stained with DAPI (4′,6-diamidino-2-phenylindole, #D9542; Sigma-Aldrich) in 1×— TBS and mounted with mowiol solution (*Amé et al., 2009*).

For β-actin IF, a classic protocol was used. Briefly, cells were fixed with 2% PFA, 1× PBS for 10 min on ice. After three washes with 1× PBS, 0.1% Triton X-100, 1% Bovine Serum Albumin, cells were incubated with the primary antibody diluted in 1× PBS, 0.1% Triton X-100, 1% Bovine Serum Albumin, 5% normal goat serum O/N at 4°C in a humidified chamber. Slides were washed three times for 5 min at RT and incubated with secondary antibody for 1 hr at RT. After washing, cells were stained with DAPI in 1× PBS and mounted with mowiol solution.

Chromatin fiber protocol was adapted from *Sullivan (2010)*. Trypsinized HeLa cells were incubated in hypotonic buffer (75 mM KCl) for 10 min at RT, before cytospining for 10 min at 400 rpm. Slides were immersed in freshly prepared fiber lysis buffer (2.5 mM Tris HCl pH7.5, 0.5 M NaCl, 1% Triton X-100, 0.4 M urea) for 15 min at RT, then in fixation buffer (4% formaldehyde (#F8775; Sigma-Aldrich), 1X PBS, final pH 7.4) for 10 min at RT, and finally in permeabilization buffer (1X PBS; 0.1% Triton X-100) for 7 min at RT. After blocking (1X PBS, 0.5% Bovine Serum Albumin, 0.01% Triton X-100), chromatin fibers were stained O/N at 4°C in a humidified chamber with primary antibody diluted in blocking solution complemented with 1% normal goat serum. Slides were washed three times for 5 min at RT with 1X PBS, 0.05% Tween 20, before incubation with secondary antibody for 1 hr at RT. After washing, the protocol was repeated for co-IF, and the fibers were then stained with DAPI in 1× PBS and mounted with mowiol solution.

When FISH was performed, antibody protein complexes were crosslinked (8% formaldehyde diluted in distilled water) for 10 min at RT, denatured in 70% formamide (#F47670; Sigma-Aldrich), 2X SSC buffer (#46 – 020 CM, Corning, Manassas, VA) for 8 min at 78°C, and then incubated with denatured probed (tagged with biotin-16-dUTP [#11093070910, Roche, Indianapolis, IN] or Cy$^{TM}$5 dUTP [#PA55022, GE Healthcare, Pittsburgh, PA] by nick translation method) O/N at 37°C in a

humidified chamber. Slides were washed 5 min at 45°C three times with 50% formamide, 2X SSC solution, and four times with 2X SSC, 0.05% Tween 20 solution. Slides were blocked in 4X SSC, 0.1% Tween 20, 3% Bovine Serum Albumin for 30 min at RT. Following the incubation with the secondary antibody at 37°C for 1 hr, slides were washed four times for 5 min each at 45°C with 4X SSC, 0.1% Tween 20, stained with DAPI in 2X SSC, and mounted coverslip with mowiol solution.

All samples were observed with a DeltaVision RT system (Applied Precision, Issaquah, WA) controlling an interline charge-coupled device camera (Coolsnap, Roper Scientific, Martinsried, Germany) mounted on an inverted microscope (IX-70; Olympus, Center Valley, PA). Images were captured by using a 60×— at 0.2 µm z sections for cell and 100×— objective at 0.1 µm z sections for chromatin fiber, deconvolved, and projected by using softWoRx (Applied Precision). Three independent experiments were performed and in each, 5–10 chromatin fibers or 30–50 cells were analyzed per slide.

## IF analysis

To quantify IF signals, the acquisition of pictures for all samples of an experiment was performed with the same time of exposure during the same day to avoid variability from the instrument. Using ImageJ (ImageJ 1.43U), signal intensity of each CENP-A or CENP-B spot inside of the nucleus (as defined by the DAPI staining) was extracted. The background level of the nucleus was subtracted from the average value of the spot intensity per cell. For each experiment, the average value of the spot intensity per cell and the ratio signal intensity in treated condition vs signal intensity in non-treated condition was measured. The mean and standard deviation values of three experiments are presented in the *Supplementary file 1*.

## Chromatin extraction and immunoprecipitation (IP)

Five F175 flasks of HeLa cells (70–80% of confluence) were used for IP. Cells were trypsinized (#25300; Gibco) and washed three times with cold $1\times$ PBS, 0.1% Tween 20 coupled with 5 min centrifugation at 800 rpm at 4°C. Nuclei were isolated in TM2 buffer (20 mM Tris HCl pH8, 2 mM $MgCl_2$, 0.5 mM PMSF) complemented with 0.5% NP40 substitute (#74385, Sigma-Aldrich), and washed once with TM2 buffer. Chromatin was digested 2 min at 37°C with 0.2 unit/ml of MNase (#N3755; Sigma-Aldrich) in 0.1 M TE buffer (0.1 M NaCl, 10 mM Tris HCl pH8, 0.2 mM EGTA) complemented with 2 mM $CaCl_2$. The reaction was stopped by addition of 10 mM EGTA and transferred to ice. After centrifugation for 5 min at 800 rpm at 4°C, the nuclear pellet was resuspended in 1 ml low-salt buffer ($0.5\times$ PBS, 5 mM EGTA, 0.5 mM PMSF, protease inhibitor cocktail [#05892953001; Roche]), and the chromatin was extracted O/N at 4°C in an end-over-end rotator. An aliquot of the supernatant obtained after centrifugation for 5 min at 8000 rpm at 4°C was saved as input (1.5%). At 4°C, sample was precleared with protein A/G Plus agarose beads (#sc-2003; Santa Cruz Biotechnology, Dallas, TX) for 30 min at 4°C in an end-over-end rotator, incubated with the primary antibody for 4 hr, followed by IP with protein A/G Plus agarose beads for 2 hr. After centrifugation, an aliquot was saved as unbound (UB, 1.5%), and the bead-associated IP was washed three times with low-salt buffer, and stored at −20°C in Laemmli buffer (30 µl) for Western blot analysis.

## RNA-chromatin IP and RNA purification

A general scheme is presented on *Figure 4—figure supplement 3*. Five F175 flasks of eG1-synchronized HeLa cells (70–80% of confluency) were used for IP. After trypsinization, cells were washed two times with cold $1\times$ PBS, 0.1% Tween 20, fixed 10 min at RT with 1% formaldehyde, quenched by addition of 125 mM glycine, and washed twice with cold 1X PBS, 0.1% Tween 20. Samples were treated as described above (i.e., Chromatin extraction and immunoprecipitation) in presence of 10 mM Ribonucleoside Vanadyl Complex (RVC, #1402; NEB). After centrifugation 5 min at 800 rpm at 4°C following the MNase digestion, the supernatant was saved and named soluble fraction, whereas the nuclear pellet was suspended in 1 ml low-salt buffer ($0.5\times$ PBS, 5 mM EGTA, 0.5 mM PMSF, protease inhibitor cocktail, 10 mM RVC) and chromatin was extracted O/N at 4°C. The supernatant obtained after centrifugation 5 min at 8000 rpm at 4°C was named chromatin fraction. IPs were performed as described previously with both fractions (i.e., Chromatin extraction and immunoprecipitation). After washes, bead-associated IPs were divided in two equal samples for protein analysis by Western blot and for RNA study.

For RNA study, RNA protein complexes were eluted from protein A/G Plus agarose beads by incubation in an end-over-end rotator for 15 min at RT with 250 µl elution buffer (1% SDS; 0.1 M NaHCO$_3$). The supernatant was saved and the elution step was repeated once more. All samples (input, unbound and IP) were denatured at 65°C for 2 hr with 200 mM NaCl. Proteins were digested with 20 µg of proteinase K (#AM2548; Ambion) in the presence of 40 mM Tris HCl pH6.5, 10 mM EDTA at 42°C for 45 min. To avoid genomic DNA contamination, samples were treated with DNase I for 30 min at 37°C. The reaction was stopped by addition of 5 mM EDTA, and RNAs were purified by phenol:chloroform:isoamyl alcohol method and ethanol precipitation. Samples were stored at 80°C until further use in retro-transcription PCR and Northern blot.

## Western blot

Samples in Laemmli buffer were denatured 5 min at 95°C, plunged on ice for 2 min, loaded into a 4 20% SDS-PAGE (#456 – 1093; Biorad, Hercules, CA) for separation in 1×— Tris Glycine SDS Running Buffer (#161 – 0732; Biorad), and transferred to Whatman nitrocellulose membrane (#10439396; Sigma-Aldrich) in Tris Glycine transfer buffer (#351-087-131; Quality Biological) diluted in 20% ethanol. Membrane was blocked in Odyssey blocking buffer (#927 – 40000; Li-Cor, Lincoln, NE) diluted in 1× PBS (1:1) at RT for 1 hr, and incubated with primary antibody diluted in blocking buffer complemented with 0.1% Tween 20 O/N at 4°C. After three washes in 1× PBS, 0.1% Tween 20, the membrane was incubated with the secondary antibody conjugated to IRDye680 (#926 – 68072 and #926 – 68073; Li-Cor) diluted in blocking buffer complemented with 0.1% Tween 20% and 0.05% SDS for 1 hr at RT. The membrane was washed in the same conditions than previously and proteins were detected by scan on Odyssey CLx scanner (Li-Cor).

## Protein quantification

eG1-synchronized HeLa cells treated with α-amanitin (as described in Cell culture and RNA Polymerase inhibition) were resuspended in lysis buffer (20 mM Tris HCl pH7.5, 400 mM NaCl, 2 mM dithiothreitol, 1% Nonidet P40 substitute, 0.5 mM PMSF, protease inhibitor cocktail). After three cycles of freezing and thawing, extract was centrifuged for 20 min at 12,000 rpm at 4°C. The cleared suspension was quantified by UV spectroscopy, and 50 µg of proteins were resuspended in Laemmli buffer. After separation into a 4 20% SDS-PAGE, transfer to a nitrocellulose membrane and incubation with primary and secondary antibodies (as described in Western Blot), proteins were detected by scan on Odyssey CLx scanner and quantified with ImageStudioLite software (Li-Cor). For each experiment, the ratio of signal intensity in treated condition vs signal intensity in non-treated condition was measured. The means and standard deviations of three experiments are presented in the figure.

## Northern blot

Northern blotting protocol was adapted from *Summer et al. (2009)* and http://archive.bio.ed.ac.uk/ ribosys/protocols/website_Northern_blotting.pdf. 5 µg of Trizol extracted RNAs or 1 µg of immunoprecipitated RNAs was separated on 4% urea PAGE against 0.5×— TBE buffer at 25 W for 90 min. RNAs were transferred to Amersham Hybond-NX membrane (#RPN203T; GE Healthcare) for 2 hr at 65 V, UV-cross-linked, blocked for 1 hr in SES buffer (0.5 M Na$_3$PO$_4$ pH7.2, 7% [wt/vol] SDS, 1 mM EDTA), and hybridized O/N at 37°C with radiolabelled α-satellite probes (end-labeling method using primer extension-system AMV reverse transcriptase kit, #E3030; Promega, Madison, WI) diluted in SES buffer. Membrane was washed in 6×— SSPE (1.08 M NaCl, 0.06 M NaH$_2$PO$_4$, 20 mM EDTA, pH adjusted to 7.4) two times for 15 min at 37°C and two times for 30 min at 42°C. Blot was exposed to P32-sensitive film (Hyblot CL film, #E3012; Denville, Saint Laurent, Canada) at −80°C to reveal the potential interaction for a short (<24 hr) or long (>24 hr) period of time. The sequences of the radiolabeled probes are indicated in *Supplementary file 10*.

## Computational analysis

Computational prediction of RNA binding residues was performed with BindN$^+$ program (http://bioinfo.ggc.org/bindn+/) with a specificity equal to 85% (*Wang et al., 2010*). We used human CENP-A (P49450), H3.1 (P68431) and HJURP (Q8NCD3), and Scm3 (Q12334) protein sequences from uniprot database.

## Transfection with shRNA

pGFP-V-RS plasmids expressing shRNA sequence were purchased from Origine (Rockville, MD). Two controls were used for each experiment: the empty vector (#TR30007) and the vector expressing scrambled sequence cassette (#TR30013; shRNA$^{scram}$ 5′-GCACTACCAGAGCTAACTCAGATAGTACT-3′). Two shRNA sequence cassettes were designed from the centromeric α-satellite consensus sequence (*Waye and Willard, 1987*): shRNA$^{sat1}$ 5′-TGTGTGCATTCAACTCACAGAGTTG-3′ and shRNA$^{sat2}$ 5′-CAACTCACAGAGTTGAACCTTCCTT-3′ (*Figure 6—figure supplement 1*); and from the cenRNA#1 sequence (*Figure 5—figure supplement 1*): shRNA$^{cenRNA\#1}$ 5′-TGCTAGACAGCCAATGCAATTCCTCATTA-3′.

Cells were transfected with Escort II Transfection Reagent (#L6037; Sigma-Aldrich) following manufacturer instruction. 48 hr after transfection, the medium was replaced every 2 days with fresh medium complemented with 0.5 µg/ml puromycin (#A1113802; Gibco) to select transfected cells. At day 6, cells were either treated for IF or RNA extraction.

## Quantitative PCR (qtPCR)

To detect α-satellite expression level in shRNA-transfected cells, RNAs were extracted, quantified by UV-spectroscopy, and equal quantities were retro-transcribed using Superscript III First-Strand Synthesis kit as described above (i.e., RNA extraction, retro-transcription, and Polymerase Chain Reaction). To perform qtPCR, complementary DNAs (cDNAs) samples were prepared using the iQ SYBR Green supermix (#170–8880; Biorad) following manufacturer's protocol. Control reactions without the cDNA were performed to rule out non-specific amplification. The qtPCR was run on Step one plus Real time PCR system (Applied Biosystem, Grand Island, NY). Primer sequences are available on *Supplementary file 3*.

The comparative cycle threshold ($C_T$) method was used to analyze the expression level of α-satellite transcripts. $C_T$ values were normalized against the average $C_T$ value of the housekeeping gene β-actin. The $\Delta\Delta C_T$ values were determined from the scrambled shRNA samples. Relative fold differences ($2^{-\Delta\Delta CT}$) are indicated on figure.

## β-galactosidase assay

Senescent cells were detected using the protocol developed by *Itahana et al. (2007)*. Briefly, 6 days post transfection with shRNA, HeLa cells grown in a six-well plate were washed two times in 1× PBS, fixed 5 min at RT with 3.7% formaldehyde in 1X PBS, and washed twice with 1X PBS. Cells were stained with the X-gal staining solution (1 mg/ml of X-gal [#B9146; Sigma-Aldrich], 40 mM citric acid/sodium phosphate buffer pH 6.0, 5 mM potassium ferricyanide [#702587; Sigma-Aldrich], 5 mM potassium ferrocyanide [#P3289; Sigma-Aldrich], 150 mM NaCl, 2 mM MgCl$_2$) O/N at 37°C. After rinsing, cells were observed under a light microscope for blue color, indicator of senescence.

## Statistical analysis

Standard deviation was determined for all quantification measures. To test the significance of these measures, a two-tailed, paired Student's *t* test was performed. For all tests α was assumed to be 0.05. The p-value is indicated on the figures or tables each time it was evaluated.

## Acknowledgements

We thank Shiv Grewal (LBMB, NCI), Tom Misteli (LRBGE, NCI), Rachel O'Neill, and members of our lab for thoughtful discussion and critical feedback; Trent Bowen and Rajbir Gill for technical assistance; David Sturgill for assistance with computational analysis; and James McNally (LRBGE, NCI) and Kathy McKinnon (Vaccine Branch, NCI) for the access to microscopy facility and FACS core facility, respectively. YD and DQ are supported by the Intramural Research Program of the Center for Cancer Research at the National Cancer Institute/NIH. Dr. David Sturgill (staff bioinformatician, LRBGE, NCI) discovered the correct identity of cenRNA1 by performing the extensive analysis discussed in the correction, and Dr. Minh Bui (lab biologist, LRBGE, NCI) performed rigorous ASO/LNA analysis to understand why the cenRNA1 knockdown yielded a mitotic defect observed in the original manuscript.

## Additional information

### Funding

| Funder | Grant reference number | Author |
| --- | --- | --- |
| National Cancer Institute | Center for Cancer Research, Intramural Research Program | Yamini Dalal |

The funders had no role in study design, data collection and interpretation, or the decision to submit the work for publication.

### Author contributions

Delphine Quénet, Conception and design, Acquisition of data, Analysis and interpretation of data, Drafting or revising the article, Contributed unpublished essential data or reagents; Yamini Dalal, Conception and design, Analysis and interpretation of data, Drafting or revising the article

### Decision letter and Author response

Decision letter https://doi.org/10.7554/eLife.03254.036
Author response https://doi.org/10.7554/eLife.03254.037

## Additional files

### Supplementary files

• Supplementary file 1. RNA Polymerase II inhibition results in CENP-A loss at centromere at early G1. eG1 synchronized cells were treated 2 hr with or without α-amanitin and stained for centromeric protein CENP-A or CENP-B. After image acquisition, immunofluorescent signals were quantified using ImageJ.
DOI: https://doi.org/10.7554/eLife.03254.022

• Supplementary file 2. Alignment of cenRNA#1 28bp repeat to CENP-B box
DOI: https://doi.org/10.7554/eLife.03254.028

• Supplementary file 3. Best alignment hits for cenRNA#1 regions without contiguous full-length adapter sequences.
DOI: https://doi.org/10.7554/eLife.03254.029

• Supplementary file 4. List of databases and sequences questioned to identify cenRNA#1 origin.
DOI: https://doi.org/10.7554/eLife.03254.030

• Supplementary file 5. Best alignment hits for cenRNA#1 regions without contiguous full-length adapter sequences.
DOI: https://doi.org/10.7554/eLife.03254.031

• Supplementary file 6. Changes to cenRNA#1 shRNA sequence.
DOI: https://doi.org/10.7554/eLife.03254.032

• Supplementary file 7. Alignment of cenRNA#1 shRNA sequence.
DOI: https://doi.org/10.7554/eLife.03254.033

• Supplementary file 8. List of antibodies used in this study.
DOI: https://doi.org/10.7554/eLife.03254.023

• Supplementary file 9. List of LNA ASO sequences and LNA probes..
DOI: https://doi.org/10.7554/eLife.03254.034

• Supplementary file 10. List of primer sequences used in this study.
DOI: https://doi.org/10.7554/eLife.03254.024

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
