## [Decision Letter]

Your manuscript titled, A centromeric long non-coding RNA is required for CENP-A targeting to the human centromere, was reviewed by three experts in the field and by a member of the Board of Reviewing Editors (BRE). After a full discussion of the study and the reviews, I am happy to report that the reviewers and the BRE member found the study of great interest to the journal, and therefore, we are happy to consider a revised manuscript addressing the following issues. I have also included the full-reviewers' comments at the bottom of this letter, which contain minor issues and textual modifications that need to be addressed in the revised manuscript.

1) Your report of the 665 bp portion of the 1.3 kb sequence lacks alpha satellite, even though it was picked up on a Northern blot using alpha satellite as probes. Please provide more details concerning the probe and also repeat a Northern that is reprobed with this 665-bp sequence to determine whether the sequence identified is actually present in the RNA. This point needs to be clarified for our reviewers and readership.

2) Provide negative and positive controls for the experiments as suggested by all three reviewers (the detailed reviews are included below).

3) Provide quantification for the results for Figures 1a, 3c, 4c, and 6.

4) Identify the genomic region from where the centromeric RNA originates. One reviewer has suggested that you could clone the full length RNA, and validate more carefully its implication in CENP-A deposition (further information regarding the reviewer's suggestion is included below).

Reviewer #1

How typical centromeres, such as those of humans, are targeted for function remains poorly understood. Whereas it might be assumed that alpha satellite is necessary and sufficient, because all human centromeres are embedded in alpha satellite, the occurrence of neocentromeres implies that alpha satellite is not sufficient for localization of the centromere foundation protein, the histone H3 variant CENP-A. A chaperone protein, HJURP, has been identified as being a requirement, and it is known that at least some alpha satellite sequences are transcribed, but these finding do not provide a clue as to how centromeres are maintained throughout the cell cycle. Now the authors show that transcription during early G1 is necessary for correct localization of CENP-A and describe an RNA sequence component of the CENP-A assembly complex that is required for normal centromere function. The data supporting these major novel points are convincing, making this study an exciting advance in understanding how centromeres are maintained.

Although the importance of the central findings justify publication in *eLife*, there were some issues that need to be addressed.

1) The authors report that a search for a 665 bp portion of the 1.3 kb sequence failed to find it in the human or any other genome. Oddly, this portion lacks alpha satellite, even though it was picked up on a Northern blot using alpha satellite as probe. Some details are needed concerning exactly what the probe was. It would seem surprising if the 665-bp sequence is also absent from human short-read archives, as there is no reason to expect there to be a bias against a centromere-linked sequence in unassembled short reads produced from whole genomic DNA. If the sequence is not also found there, then some rational explanation for the phantom nature of this sequence would be welcome. For example, perhaps it derives from an as-yet unidentified endogenous RNA virus particle. In any case, the Northern should be reprobed with this 665-bp sequence to determine whether the sequence identified is actually present in the RNA.

2) Figure 1B: All 3 IPs show enrichment relative to Input and Unbound. A negative control is needed for this experiment to verify that some of the enrichment isn't a technical artifact, such as background chromatin preferentially sticking to beads or some other systematic bias.

Reviewer #2

This manuscript presents an interesting idea, namely that a long noncoding RNA associates with HJURP and CENP-A and facilitates CENPa nucleosome assembly in early G1. I think the idea is consistent with the data, but I think there are many aspects of the experiments that could be significantly improved to be more convincing. In general, the experiments need more controls and quantification to be interpretable.

1) Its not clear if the shRNAs actually target the novel RNA sequence that the authors identified, they target a-sat which colocalizes with the novel RNA sequence. This seems like a crucial piece of data. Why not design a shRNA to knock down the identified RNA sequence, show it is depleted by northern, and then perform the functional assays? The significance of the sequence identified is otherwise unclear.

2) No negative controls for the western blots in Figure 1b. Also, there are multiple bands detected for CENP-C and TBP, which bands are the relevant bands?

3) No positive control for Figure 4figure supplement 1B (-RNAse).

4) Background is too high in mock IP Figure 4figure supplement 4 to know if there is any significant pull down in the mock IPs. There is a lot of smearing in part B that does not seem to be as prevalent as Figure 4 S1C.

5) In Figure 4B there is no mock control.

6) A lot is riding on the chromatin fiber experiments. In the figure legends, there are indications of repeats and number of fibers, but there is no quantification of the results for Figures 1a, 3c, 4c, and 6. Colocalization is very difficult/impossible to see in the figures as presented.

Reviewer #3

In this study, Quenet and Dalal investigated the mechanism of human CENP-A deposition at centromeres during early G1 phase of the cell cycle. Using immuno-precipitation and chromatin fibers imaging, they were able to identify a single 1.3 kb centromeric RNA that is claimed to play a critical role in CENP-A targeting and deposition at centromeres. The identified RNA molecule interacts with the soluble pre-assembly HJURP and CENP-A complexes in vivo. Its down-regulation is suggested to result in the loss of CENP-A and HJURP from centromeres and to subsequent chromosome instability. The 1.3 kb molecule is transcribed by RNA Polymerase II from centromeric DNA. Accordingly, inhibition of RNA Polymerase II transcription abrogates the recruitment of both CENP-A and its chaperone HJURP to centromeres.

Overall, I think this work provides some interesting observation but they are too preliminary and contradictory to deserve publication in the present form. The authors need to seriously revise the paper, strengthen the biochemistry experiments and to validate their model in vivo and in vitro.

The manuscript in its present form exhibits several caveats. The most important issues are listed below:

1) Figure 1B, the Western Blot. The signal with the anti-CENP-C antibody is very weak (just slightly above the background). The authors should present a more convincing blot.

2) Mounting evidence supports the hypothesis that an RNA component is involved in centromere function; although how RNAs participate in centromere formation in mammals has remained unknown. This study identifies a novel RNA component but fail to provide convincing evidence for its implication in CENP-A deposition. The proposed model for RNA guided CENP-A deposition is interesting but the identified 1.3 kb RNA molecule is very questionable. In Figure 4B, the size of the RNA identified in the HJURP complex (1.3 Kb) is different from the one identified in the CENP-A complex (1.1 Kb). How do the authors explain this discrepancy?

3) To be convincing, the authors need to identify the genomic region from where this centromeric RNA originates. They should clone the full length RNA and validate more carefully its implication in CENP-A deposition. The two shRNAs presented in Figure 5 are useless since they were designed against the ?-satellite RNA and not against the 1.3 Kb RNA identified in this study. What is the relationship between ?-satellite RNA and the 1.3 Kb RNA? Why the authors are not able to detect the association of CENP-A with ?-satellite RNA as previously described by others (Ferri et al. Nucleic Acids Res. 2009 Aug;37(15):5071-80 ; Wong et al., Genome Res. 2007;17:1146-1160).

4) Figure 4figure supplement 2 predicts the presence of several RNA binding motifs in HJURP and CENP-A but fail to provide a single experimental evidence for such interaction. The authors should reconstitute this interaction in vitro and map the interaction motifs either in CENP-A or HJURP. They should next mutate these motifs and validate their functional relevance in vivo.

5) Is it possible that the overexpression of the 1.3 kb RNA affects the centromere function and CENP-A deposition?

---

## [Author Response]

We thank the senior editor and all 3 reviewers for their positive comments and excellent experimental suggestions, both of which helped us strengthen the conclusions of our manuscript, namely that cell-cycle regulated non-coding 1.3kb centromeric RNAs participate in targeting HJURP and CENP-A to the human centromere, and that one such CENP-A-associated lncRNA (cenRNA#1) is important for cell integrity.

Reviewer #1

[] Although the importance of the central findings justify publication in eLife, there were some issues that need to be addressed.1) The authors report that a search for a 665 bp portion of the 1.3 kb sequence failed to find it in the human or any other genome. Oddly, this portion lacks alpha satellite, even though it was picked up on a Northern blot using alpha satellite as probe. Some details are needed concerning exactly what the probe was. It would seem surprising if the 665-bp sequence is also absent from human short-read archives, as there is no reason to expect there to be a bias against a centromere-linked sequence in unassembled short reads produced from whole genomic DNA. If the sequence is not also found there, then some rational explanation for the phantom nature of this sequence would be welcome. For example, perhaps it derives from an as-yet unidentified endogenous RNA virus particle. In any case, the Northern should be reprobed with this 665-bp sequence to determine whether the sequence identified is actually present in the RNA.

The reviewer has a valid point. We note that DNA FISH experiments (Figure 5B) demonstrate the centromeric origin of cenRNA#1, which we originally cloned from CENP-A IPs. The logical assumption we made is that this transcript is one of the 1.3 kb RNA species observed by Northern blot using centromeric ?-satellite probe (Figure 4B).

However as the reviewer notes, the identified CenRNA#1 is curious because no significant alignment to the current human genome build can be detected (nor to any other DNA sequences extant in the literature). Similarity to ?-satellite and chromosome X-HOR is just under the 60% cutoff used in Hasson et al. (2013) (53 and 55%, respectively). However, a weak similarity to canonical CENP-B box has been observed within this sequence for four periodically spaced 28-bp repeats (data not shown). Thus, the lack of robust identification of this sequence relative to known centromeric sequences remains a major puzzle.

There are several possibilities that might contribute to this issue:

i) Low frequency centromeric repeats from which cenRNA#1 arises were either not enriched efficiently in the Hasson et al study (peak calling relies on tag densities above a certain cutoff, and different CENP-A antibodies have variable specificities (*Gill, Walkiewicz et al, in revision*));

ii) As the reviewer suggests, this RNA arises from currently unidentified low frequency centromeric sequences.

iii) The RNA is spliced, but we think this unlikely.

iv) The RNA is unstable, making full cloning technically challenging .

We probed Northern blots with the cenRNA#1 sequence, and despite our best efforts were unable to obtain a robust signal. This supports the possibility that full length cenRNA#1 is either at low abundance or is unstable. Instability in the full-length construct could also explain why alpha-satellite consensus sequence (through which the cenRNA was cloned) are lacking in the 665bp. Thus, we specifically state in the main manuscript that we were unable to detect this RNA using Northern blots.

Importantly, reviewer comments prompted us to directly assay the functional biological relevance of cenRNA#1 by performing a targeted knockdown using shRNA^CEN#1^ construct (Figure 5figure supplement 1). Reassuringly, after shRNA^CEN#1^treatment, cells display far lower density (reduced by 70%), and abnormal morphology, with very few mitotics relative to control shRNA (Figure 5figure supplement 2). This phenotype is reminiscent to cells depleted of all centromeric ?-satellite transcripts (Figure 6). Combined with the DNA FISH experiments (Figure 5), this new piece of data lends support to the interpretation that cenRNA#1 is indeed involved in centromeric integrity. Further genetic and biochemical characterization of cenRNA#1, and other cenRNAs is currently a major effort in the lab.

2)Figure 1B: All 3 IPs show enrichment relative to Input and Unbound. A negative control is needed for this experiment to verify that some of the enrichment isn't a technical artifact, such as background chromatin preferentially sticking to beads or some other systematic bias.

We agree and apologize for leaving it out in the original version. A mock IP control showing the specificity of the three immuno-precipitations in Figure 1B, is presented in Figure 1figure supplement 2C.

Reviewer #2

This manuscript presents an interesting idea, namely that a long noncoding RNA associates with HJURP and CENP-A and facilitates CENPa nucleosome assembly in early G1. I think the idea is consistent with the data, but I think there are many aspects of the experiments that could be significantly improved to be more convincing. In general, the experiments need more controls and quantification to be interpretable.

We thank the reviewer for finding the data support our interpretation that ncRNAs are involved in CENP-A/HJURP targeting.

1) Its not clear if the shRNAs actually target the novel RNA sequence that the authors identified, they target a-sat which colocalizes with the novel RNA sequence. This seems like a crucial piece of data. Why not design a shRNA to knock down the identified RNA sequence, show it is depleted by northern, and then perform the functional assays? The significance of the sequence identified is otherwise unclear.

We agree this is an excellent experiment. An shRNA designed specifically against cenRNA#1 (Figure 5figure supplement 1) can be seen to strongly affects cell viability and morphology (Figure 5figure supplement 2), 70% lower cell density and abnormal cell morphology, as well as very few mitotics. These phenotypes are similar to those we obtained with the total centromeric ?-satellite RNA knockdown. Further biochemical and genetic characterization of cenRNA#1 (as well as other cenRNAs) is currently underway, but outside the scope of this manuscript.

2) No negative controls for the western blots inFigure 1b. Also, there are multiple bands detected for CENP-C and TBP, which bands are the relevant bands?

To test the specificity of the immuno-precipitations in Figure 1B, we present a mock IP control in which the different proteins involved in this study were examined (CENP-A, TBP, CENP-C and RNAPII). As can be seen, these proteins are observed in input and unbound, but not in the mock IP.

To detect TBP protein on Western Blot, we used the Abcam TBP antibody (ab63766), which, according the manufacturer reveals two bands- the lower band (?45kDa) is TBP, whereas the upper band (?50kDa) has an unknown identity.

A perfect antibody to reveal endogenous CENP-C protein on Western Blot does not currently exist, a problem known to plague the field. However, a band ?140kDa has been observed with several CENP-C antibodies including the Santa Cruz antibody (sc11285) presented in Figure 1B. Thus, the expected band for CENP-C is the lower band on each blot presented.

3) No positive control forFigure 4figure supplement 1B(-RNAse).

In the initial version of the manuscript, Figure 4figure supplement 1B only presented the RNA sample treated with RNAse. This experiment was performed with the same RNA sample used in Figure 4, but treated with RNAse before loading on a separate gel. This choice of running two different gels was to avoid the possibility that RNAse may diffuse and degrade RNAs loaded in neighboring wells. We performed new Northern Blot and loaded both samples on the same gel. The result is presented on Figure 4figure supplement 1B.

4) Background is too high in mock IPFigure 4figure supplement 4to know if there is any significant pull down in the mock IPs. There is a lot of smearing in part B that does not seem to be as prevalent asFigure 4S1C.

The HJURP Western blot for the soluble fraction mock control in Figure 4figure supplement 4A was changed, for a new one with a lower background. However, we constantly observe this higher background in the IP lane after incubation with HJURP antibody. We hope that this new blot will convince Reviewer 2.

There is also this smearing signal in the Northern blot for the mock control. We were unsuccessful to have a better blot to present in the new version of the manuscript, as the quantity of DNA extracted after IP is low and we performed the experiment with 1?g of RNA.

5) InFigure 4Bthere is no mock control.

The mock control for Figure 4B is presented in Figure 4figure supplement 4B. This control shows the absence of 1.3kb RNA molecule in IP lane when CENP-A or HJURP are not immuno-precipitated.

6) A lot is riding on the chromatin fiber experiments. In the figure legends, there are indications of repeats and number of fibers, but there is no quantification of the results forFigures 1a, 3c, 4c, and 6. Colocalization is very difficult/impossible to see in the figures as presented.

Optical chromatin fiber mapping experiments constitute a powerful technique to examine the localization of proteins at the chromatin level. However, quantification of the immuno-fluorescent signal on chromatin fibers is unlikely to be informative as the signal intensity often varies from one experiment to another. In our view, the information gained from such experiments is qualitative, providing a binary observation of the presence/absence or the distribution of the studied protein(s), not absolute amount (e.g., Sullivan & Karpen 2004; Sullivan et al. 2011). This is why we quantified CENP-A and CENP-B signals on whole cells in Figure 3, and not on chromatin fibers. We suggest a more meaningful measure is the number of analyzed centromeric chromatin fibers that display a particular phenotype, thus yielding a quantifiable index of the robustness of the given observation. These data are added on each figure legend presenting chromatin fibers (Figures 1A, 3C, 5A, 5B, 7, Figure 1figure supplement 2A).

Reviewer #3

[] Overall, I think this work provides some interesting observation but they are too preliminary and contradictory to deserve publication in the present form. The authors need to seriously revise the paper, strengthen the biochemistry experiments and to validate their model in vivo and in vitro.The manuscript in its present form exhibits several caveats. The most important issues are listed below:1)Figure 1B, the Western Blot. The signal with the anti-CENP-C antibody is very weak (just slightly above the background). The authors should present a more convincing blot.

Several CENP-C antibodies were tested, but none of them gave us a better result than the one presented in Figure 1B. Although the CENP-C signal is admittedly, weak, this result is reproducible, and therefore we are convinced by our interpretation.

2) Mounting evidence supports the hypothesis that an RNA component is involved in centromere function; although how RNAs participate in centromere formation in mammals has remained unknown. This study identifies a novel RNA component but fail to provide convincing evidence for its implication in CENP-A deposition. The proposed model for RNA guided CENP-A deposition is interesting but the identified 1.3 kb RNA molecule is very questionable. InFigure 4B, the size of the RNA identified in the HJURP complex (1.3 Kb) is different from the one identified in the CENP-A complex (1.1 Kb). How do the authors explain this discrepancy?

In Figure 4B, the observed difference in size results from a smiling migration, that is evident upon looking at the standard ladder at the far right (three ladders were used in this experiment, and we cropped out the one to the right in the initial version of the manuscript). Thus, there is no discrepancy in the size of the RNA identified in the HJURP complex and in the CENP-A complex. Both are ?1.3 kb long.

3) To be convincing, the authors need to identify the genomic region from where this centromeric RNA originates. They should clone the full length RNA and validate more carefully its implication in CENP-A deposition. The two shRNAs presented inFigure 5are useless since they were designed against the ?-satellite RNA and not against the 1.3 Kb RNA identified in this study. What is the relationship between ?-satellite RNA and the 1.3 Kb RNA? Why the authors are not able to detect the association of CENP-A with ?-satellite RNA as previously described by others (Ferri et al. Nucleic Acids Res. 2009 Aug;37(15):5071-80 ; Wong et al., Genome Res. 2007;17:1146-1160).

Using RNA-immuno-precipitation followed by Northern blot, we revealed an interaction between CENP-A and 1.3kb RNAs which are detected by centromeric ?-satellite primers. These data suggest that in human cells this histone variant binds ?-satellite RNAs whose size is 1.3 kb, as previously described by Ferri *et al.* in mouse cells (Nucleic Acids Research, 2009). To the best of our knowledge, Wong et al. (Genome Research, 2007) did not revealed an association of CENP-A and ?-satellite RNA, but instead used CENPC, INCENP and ?-satellite transcript. We did not test this possibility in our model, as we focused on the role of centromeric transcript in CENP-A and HJURP localization at centromeres.

Figures 5 and 7 show the importance of these centromeric ?-satellite transcripts in the targeting of CENP-A and HJURP, as their down-regulation affects the centromeric localization of these two proteins. These data are key results, because they show, for the first time, the importance of centromeric RNAs in CENP-A and HJURP targeting.

In order to better characterize these transcripts, we sequenced RNAs co-purified with CENP-A, and identified one sequence (cenRNA#1). We agree with Reviewer 2 and 3 that showing it is involved in cell integrity, is important. We provide this experiment in Figure 5figure supplement 2, designing an shRNA against cenRNA#1, and observing dramatic loss of cell density (by 70%), and abnormal cell morphology. We also see the number of mitotic cells is reduced in cells transfected with shRNA^cenRNA#1^ (data not shown). Ongoing studies are focused on a complete biochemical and genetic characterization of this and other ncRNAs associated with CENP-A, but this is outside the scope of this initial first description of RNA involvement in CENP-A/HJURP deposition.

4)Figure 4figure supplement 2predicts the presence of several RNA binding motifs in HJURP and CENP-A but fail to provide a single experimental evidence for such interaction. The authors should reconstitute this interaction in vitro and map the interaction motifs either in CENP-A or HJURP. They should next mutate these motifs and validate their functional relevance in vivo.

We agree with Reviewer 3 about the importance of *in vitro* characterization of RNACENP-A and RNA-HJURP interaction. Indeed, we are very excited about these next series of experiments. However, we believe this set of experiments constitutes an obvious follow-up project, and is outside the scope of this manuscript.

5) Is it possible that the overexpression of the 1.3 kb RNA affects the centromere function and CENP-A deposition?

Several lines of evidence in the literature suggest that the expression level of centromeric RNAs in essential for the integrity of the centromere (for review, Hall et al. 2012). For instance, Bouzinba-Segard et al. (PNAS, 2006) showed the forced accumulation of the mouse centromeric repeat transcripts (120 nucleotide long) leads to chromosome missegregation. In addition, it has been shown using human artificial chromosome, the alteration of the post-translational modifications associated with centromeres, may increase or decrease the level of transcription at this chromatin domain. In both situations, the structure and function of centromere are affected (*e.g.*, chromosome missegregation, loss of centromeric proteins) (Bergmann et al., EMBO J. 2011, Bergmann et al., Chromosome Research 2012). However, the over-expression of centromeric RNAs observed after stress suggests this alteration may also be a response to environmental conditions (Jolly et al., JCB 2004, Bouzinba-Segard et al., PNAS, 2006). Indeed, the over-expression of cenRNAs is currently being analyzed in the lab, but is outside the scope of this manuscript.